# Binary Pectin-Chitosan Composites for the Uptake of Lanthanum and Yttrium Species in Aqueous Media 

**DOI:** 10.3390/mi12050478

**Published:** 2021-04-22

**Authors:** Dexu Kong, Eny Kusrini, Lee D. Wilson

**Affiliations:** 1Saskatchewan Research Council, 125-15 Innovation Boulevard, Saskatoon, SK S7N 2X8, Canada; dek593@mail.usask.ca; 2Department of Chemistry, University of Saskatchewan, 110 Science Place, Saskatoon, SK S7N 5C9, Canada; 3Department of Chemical Engineering, Faculty of Engineering, Universitas Indonesia, Kampus UI Depok, Depok 16424, Indonesia

**Keywords:** composites, chitosan-pectin, adsorption, rare earth elements, lanthanum, yttrium

## Abstract

Rare-earth elements such as lanthanum and yttrium have wide utility in high-tech applications such as permanent magnets and batteries. The use of biopolymers and their composites as adsorbents for La (III) and Y (III) ions were investigated as a means to increase the uptake capacity. Previous work has revealed that composite materials with covalent frameworks that contain biopolymers such as pectin and chitosan have secondary adsorption sites for enhanced adsorption. Herein, the maximum adsorption capacity of a 5:1 Pectin-Chitosan composite with La (III) and Y (III) was 22 mg/g and 23 mg/g, respectively. Pectin-Chitosan composites of variable composition were characterized by complementary methods: spectroscopy (FTIR, ^13^C solids NMR), TGA, and zeta potential. This work contributes to the design of covalent Pectin-Chitosan biopolymer frameworks for the controlled removal of La (III) and Y (III) from aqueous media.

## 1. Introduction

Due to the unique metallurgical, chemical, magnetic, electrical, and catalytic properties, rare-earth elements (REEs) have a wide field of application in traditional industrial sectors (metallurgy, machinery, glass, petroleum, and chemicals). REEs are employed in diverse areas of advanced materials such as phosphors, permanent magnets, batteries, and in the nuclear industry [1,2,3]. Lanthanum (La) has the atomic number of 57 among the Group 3 elements. La is a silvery-white metallic REE that is of great importance due to its manifold use in X-ray screens, lens glass, fiber optics, batteries, capacitors, magnets, catalysts for petroleum cracking, and fluorescent lamps [1]. It is the most reactive rare earth metal, with high flammability in an oxygen environment. The melting point of Lanthanum is near 1191 K and it has relatively high density (6.15 g/cm^3^) [4]. Rare-earth element catalysts like [(C_5_Me_3_)_2_LaH]_2_ are used in the synthetic rubber industry to produce CPBR (*cis*-1,4-polybutadiene rubber) with advanced anti-fatigue life, dynamic wear, and heat of formation properties [1,5]. Yttrium is the element with the atomic number of 39 in Group 3 of the periodic table. It is a silvery-metallic non-lanthanide REE that is widely used in lasers, phosphors, alloys, medical devices, and superconductors. Phosphors are the major application for high purity oxides of yttrium, for instance, yttrium oxide is used to fabricate tricolor fluorescent lamps with high luminance, rich coloration, and long life [1,5]. Because of the important applications of REEs, their recovery from natural deposits and wastewater is beneficial to the environment and sustainable growth of such technology and the economy.

Rare earth ores often contain other minerals such as fluorite, barite, calcite, quartz, and magnetite. There are several ore pretreatment techniques that include gravity separation, flotation, and magnetic separation to make the ore extractable [1,6]. Two popular methods of obtaining REEs include solvent extraction and ion-exchange methods. Currently, solvent extraction or liquid-liquid extraction processes represent the dominant methods used by industry, due to its scalable capacity and effectiveness [4]. There remain significant challenges in separating REEs during the solvent extraction process. For instance, some low-quality mineral sources contain high concentrations of iron, copper, zinc, and relatively low concentrations of REEs. Most commercial organophosphorus ligands extract iron (III) species more efficiently than REEs [1,4]. The conventional recovery of REEs from ores requires removal of such base metal ions before the solvent extraction processes. In contrast, ion exchange or solid–liquid extraction technology can recover rare earth metal ions from low-concentration ore sources with improved selectivity [7]. Commercial resins such as the Dow Chemical resin used for extracting REEs are typically prepared from poly acrylic compounds. Resin materials of this type are dominant in the ion-exchange resin market because of their low relative cost and ready accessibility from the petrochemical industry. However, there are environmental concerns related to production of these resins from petroleum-based products [3,8,9]. By contrast, the utilization of sustainable adsorbents from natural materials [10,11,12] offer an alternative to conventional synthetic resins. 

Solid-phase adsorbents derived from chitosan are viewed as environmentally friendly biosorbents for the recovery of REEs from aqueous media. When a rare earth ore is leached out as a salt-form with a strong acid, the aqueous acidic solution is enriched with a REE that will be captured by the ligands on the modified chitosan sorbent material. Once the modified chitosan biosorbent becomes saturated with metal cations, exposure to the stripping solution serves to regenerate the biosorbent. The concentrated La and Y species in the stripping solution can be isolated by precipitation [1]. Based on the reported use of chitosan as an adsorbent for metal-ion species, a working hypothesis in the present study posits that modified chitosan sorbents have greater selectivity and affinity towards La (III) and Y (III) ions, in comparison to unmodified chitosan. This research on modified chitosan is anticipated to contribute the design of improved hybrid biosorbents with greater capacity and selectivity toward REEs in aqueous media. 

To address the overall goal for the enhanced recovery of La and Y species from aqueous media, the synthesis and characterization of novel chitosan-based sorbents is proposed. To address this goal, the following objectives will be addressed herein: (i) synthesis of pectin and chitosan composites, (ii) structural characterization of the binary composites, and (iii) comparison of the adsorption properties of chitosan and its composites with La and Y cation species at various conditions. This research will contribute to the knowledge gap concerning the utilization of biopolymer composite adsorbents for the uptake of REEs, along with new insight on the role of pectin and chitosan in such types of composite materials [13,14]. These research objectives will be addressed through complementary characterization of the Pectin-Chitosan adsorbents by spectroscopy (FTIR, ^13^C solid state NMR), thermal gravimetry analysis (TGA), and adsorption isotherm studies in aqueous media. Together, the results will be shown to provide further insight on the adsorption process of La or Y ions with modified chitosan sorbents. 

## 2. Materials and Methods

### 2.1. Materials

Lanthanum (III) chloride heptahydrate, arsenazo III (calcium-sensitive dye), yttrium (III) trichloride hexahydrate, potassium chloride, sodium hydroxide, and 37% hydrochloric acid. Chitosan (Mwt. ~50,000–190,000 g/mol) with an average degree of deacetylation (DDAc) of 75–85%, dimethyl sulfoxide (>99.7%, DMSO), and pectin from citrus peel galacturonic acid ≥74.0% (dry basis) were ACS grade materials obtained from Sigma-Aldrich (Edmonton, AB, Canada).

### 2.2. Synthesis of Pectin-Chitosan Binary Composites

#### 2.2.1. Pectin-Chitosan Polyelectrolyte Complexes in Water

To prepare the 1:5 pectin/chitosan composite (PC 15 W) in water, the 2% wt. chitosan solution was prepared by dissolving chitosan (ca. 2 g) into 98 g of the aqueous acetic acid 2% *w*/*w* and the pectin solution was prepared by dissolving pectin (ca. 1.37 g) into 68.58 g of DI water to make a 2 wt.% solution. In a 150 mL beaker, 50 g of a chitosan (2 wt.%) solution was mixed with 10 g of a pectin (2 wt. %) solution at 23 °C with a magnetic stirrer at 1000 rpm overnight. The mixture was neutralized with 1M NaOH (*aq*) 12 h after the mixing step until a pH value of 6.8 resulted in a suspension of Pectin-Chitosan particles. The resulting Pectin-Chitosan composite product was filtered by a vacuum pump with Whatman 42 ashless filter paper and washed with deionized water, where the filtrate reached a low conductivity (35 µS/cm). The final products were air-dried for 48 h. The procedure for making composites at 1:1 and 5:1 weight ratios (PC 11 W and PC 51 W) was similar to that described for PC 15 W. 

#### 2.2.2. Sonication Assisted Synthesis of Pectin-Chitosan Composites in DMSO

To prepare PC 15 S, pectin (ca. 1.5 g) and chitosan (ca. 7.5 g) were dispersed in DMSO solution (200 mL). The pectin and chitosan mixture in DMSO was sonicated for 10 min. After cooling, the brown-dark Pectin-Chitosan composites were filtered, washed with DI water, and dried in the fume hood at 23 °C. The preparation of the PC 11 S and PC 51 S composites were similar to the above, except that pectin (ca. 2 g and 7.5 g) and chitosan (ca. 2 g and 1.5 g) were suspended in 200 mL DMSO solvent. 

### 2.3. Characterization of Composite Materials

Several complementary tools for the characterization of the composites were employed: thermal gravimetric analysis (TGA), Fourier Transform Infrared (FTIR) spectroscopy, solid-state ^13^C nuclear magnetic resonance (^13^C NMR) spectroscopy, zeta potential measurements (ζ-potential), and ultraviolet-visible (UV-vis) spectrophotometry. 

#### 2.3.1. TGA 

Thermogravimetry analysis (TGA) profiles was carried out using open aluminum pans with a Q50 TA (New Castle, DE, USA) instrument. The heating rate (5 °C min^−1^) profile was monitored from 30 to 500 °C, with N_2_ as the purge and thermal regulation gas. 

#### 2.3.2. FTIR Spectroscopy

Fourier transform infrared (FT-IR) spectra of powdered samples were obtained as 1 wt.% solid samples mixed with KBr and analyzed in their powder form using the diffuse reflectance mode with a BIO-RAD FTS-40 spectrophotometer (Cambridge, MA, USA). Data collection used multiple (n = 64) scans to obtain spectra with a 4 cm^−1^ resolution that was corrected against a background spectrum of spectroscopic grade KBr over a fixed spectral range (400–4000 cm^−1^).

#### 2.3.3. ^13^C solid State NMR Spectroscopy

^13^C solids NMR spectra were obtained using a Bruker AVANCE III HD spectrometer operating at 125.77 MHz with a 4 mm DOTY CP-MAS probe. The ^13^C NMR spectra employed a CP/TOSS (Cross Polarization with Total Suppression of Spinning Sidebands) pulse sequence with a spinning speed of 6 kHz. For all the samples, 5120 scans were accumulated with a recycle delay of 2 s, and chemical shifts were referenced to adamantane at 38.48 ppm (low field signal).

#### 2.3.4. Particle Size, Polydispersity Index (PDI), and Zeta Potential Measurements 

The particle size and zeta potential were measured using a Zetasizer Nano ZS model ZEN 3600 (Malvern Instruments, UK). The cuvette is a polystyrene latex material with a refractive index 1.59 and an absorption of 0.01. Chitosan, pectin, and the composites were suspended in DI water at pH 6.5, respectively, where the sample aliquots of supernatant were used for analysis. Particle size (hydrodynamic radius was estimated by dynamic light scattering. This instrument determines the particle size distribution by measuring the intensity fluctuations over time of a laser beam (λ = 633 nm) scattered by the sample at an angle of 173°. Zeta potential (ζ) measurements were executed based on laser Doppler anemometry, using the same instrument noted above. 

#### 2.3.5. Uptake of Y (III) and La (III) by Pectin and Chitosan Composites

The adsorption properties of the samples were evaluated using Y (III) and La (III) cations in a batch mode process. About 10 mg of pectin and chitosan composites were respectively added into vials that contained 10 mL of Y (III) or La (III) solution at a variable concentration (10, 20, 40, 60, 80, 100, 120, 140, 160, and 180 mg/L). The vials were mixed using a horizontal shaker (SCILOGEX SK-O330-Pro) for 24 h at 23 °C and 150 rpm. After reaching equilibrium, the samples were centrifuged and the supernatant was sampled with needle syringes. The optical absorption of Y (III) and La (III) were determined using the UV-Vis spectrophotometer at λ_max_ = 655 nm for Y (III) and λ_max_ = 650 nm for La (III) by formation of a complex with arsenazo (III) in aqueous solution [5]. Standard stock solutions (1000 mg/L) were prepared by dissolving the required amount of yttrium (III) chloride hexahydrate in DI water. Calibrant standard solutions of Y (III) were prepared from 1 mg/L to 5 mg/L KCl/HCl buffer solution having pH 2 was prepared by mixing 8.1 mL of 0.2 M HCl and 41.9 mL of 0.2 M KCl solution and diluted to 100 mL with DI water. The pH adjustment was done by using 0.01 M NaOH solution. The 0.1% arsenazo (III) solution was made by dissolving 10 mg of the reagent in 10 mL of DI water. To determine the Y (III) concentration in calibration solutions and sample solutions, 1.0 mL of standard solution or sample solution mixed with 0.2 mL of arsenazo (III) solution and 1.0 mL (pH 2) buffer solution. The mixture was diluted to 5.0 mL with DI water and measured at 655 nm for Y (III). A similar method used for determining the concentration of La (III) was employed. Two common adsorption isotherms (Langmuir and Freundlich models) were used to study the uptake of La (III) and Y (III) with Pectin-Chitosan composites (cf. Equations (1) and (2)). The Langmuir model assumes monolayer adsorption with a finite number of binding sites that are homogeneous in nature on the adsorbent surface, where no interaction occurs between adsorbed species, as described by Equation (1):(1)$$Q_{e}=\frac{Q_{m}K_{L}C_{e}}{\left(1+K_{L}C_{e}\right)}$$*Q_m_* (mg/g) is the maximum monolayer adsorption capacity of the adsorbate bound onto adsorbent, *Q_e_* (mg/g) is the amount of the adsorbate bound at equilibrium, *C_e_* (mg/L) is the unbound adsorbate concentration in solution at equilibrium, and *K_L_* (L/mg) is the Langmuir adsorption constant. By comparison, the Freundlich model resembles the Langmuir model, except that it assumes that the sorbent has a heterogeneous surface with nonequivalent binding sites and variable enthalpy of adsorption, as described by Equation (2):(2)$$Q_{e}=K_{f}C_{e}^{\frac{1}{n}}$$*Q_e_* (mg/g) is the amount of the adsorbate bound onto the adsorbent at equilibrium, while *K_f_* (L/g) and n are the Freundlich adsorption constants for a given adsorbent-adsorbate system at specific conditions.

## 3. Results and Discussion

As noted above, several types of binary Pectin-Chitosan adsorbent materials were prepared herein according to variable synthetic conditions using adapted methods that are reported elsewhere [10]. The characterization of the materials and selected physicochemical properties rely on various complementary methods: FTIR/^13^C solid-state NMR spectroscopy, TGA, zeta potential, and the adsorption properties in aqueous media toward La (III) and Y (III) ions. The results for the structural and physicochemical characterization of the composite materials are outlined in the sections below.

### 3.1. TGA Results for the Pectin, Chitosan, and Pectin-Chitosan Composites

The TGA profiles were used to study the thermal stability of chitosan, pectin, and the binary composites. Chitosan has two weight lost events; one centered ca. 120 °C that ranged from 50 °C to 280 °C with about 4% weight reduction due to loss of moisture. The second event was centered ca. 300 °C that extended up to 400 °C to yield a weight loss more than 80% that related to thermal decomposition of chitosan [13]. Pectin had two thermal events, the first transition was centered ca. 80 °C and ranged from 40 °C to 200 °C due to loss of bound water. The second event was centered at 225 °C that ranged from 200 °C to 300 °C due to the decomposition of pectin, in agreement with another study [15]. Materials prepared in water were anticipated to favor the formation of polyelectrolyte complexes (PECs) due to the higher dielectric constant of water that favors acid–base reactions over amide bond formation. Pectin-chitosan composites prepared in water such as PC 51 W reveal lesser weight loss before 180 °C and thermal events at 235 °C and 280 °C, which reveal features noted for pristine pectin (ca. 235 °C) and the higher thermal stability event (ca. 280 °C) consistent with PEC formation. The latter feature supports polyelectrolyte complex formation since its thermal stability exceeds that of the parent biopolymers. The results concur with the role of ion–ion interactions between the biopolymer chains [14]. By contrast, the covalent biopolymer (PC 51 S) contains amide linkages between the pectin and chitosan biopolymers, which confers unique thermal stability to the composite prepared in DMSO over its counterpart in water (PC 51 W) [13]. In Figure 1, PC 51 S has a higher decomposition temperature at 300 °C, in agreement with its framework structure that contains covalent cross-linked amide linkages [13,16]. The higher decomposition temperature of PC 51 S is due to the covalent amide linkages versus the PEC network for PC 51W.

### 3.2. FTIR Spectral Results

The FTIR spectra of pectin, chitosan, and their PC 51 S and PC 51 W composites are shown in Figure 2a, whereas the spectra for PC 51 S are shown in Figure 2b, before and after the adsorption process with La (III).

The broad IR band at 1600 cm^−1^ for chitosan (cf. Figure 2a) relates to the N–H bending of a primary amine group of the glucosamine units. Pectin reveals a strong intensity stretching band (C=O) from non-ionized carboxy groups (-COOH and -COOCH_3_) of galacturonic acid at 1750 cm^−1^, and lower intensity bands for the symmetric and anti-symmetric carboxylate (-COO−) vibration at 1442 and 1673 cm^−1^ [17,18]. The increased sharpness of the band at 1595 cm^−1^ shows an increase of the amide II band (N-H) for PC 51 S. Comparing the IR spectra of PC 51 S and PC 51 W reveals differences in the N-H signatures for the presence of amide linkages for PC 51 S [19]. The broad vibrational bands at 2850 and 3300 cm^−1^ correspond to alkyl (-CH) and hydroxyl (-OH) stretching vibrations of PC 51 S. The FTIR results in Figure 2b show a decreased signal intensity for the OH groups in the composite PC 51 S after La (III) adsorption [20]. The observed changes in the -OH band highlight the role of such groups as active adsorption sites of the composite that bind with the metal ions. The results are consistent with other reports that indicate the key role of the -COOH groups of pectin for adsorption of REEs [3]. 

### 3.3. ^13^C Solid-State NMR Spectral Results

The chitosan material in this study has ca. 25% acetyl groups due to partial deacetylation of chitin. The ^13^C NMR lines at 23 and 173 ppm relate to the methyl and carbonyl groups due to the presence of acetyl groups at C2 in chitosan. Other chemical shifts correspond to different ^13^C groups of chitosan as shown in Figure 3a and Figure 4 [21,22]: C1 (104 ppm), C2 (57 ppm), C3 and C5 (75 ppm), C4 (82 ppm), C6 (61 ppm), C7 (ppm), and C8 (ppm). The chemical shifts reported for chitosan concur with assignments reported in other studies [23,24].

The molecular structure of pectin is shown in Figure 3b. The NMR spectrum of pectin in Figure 4 has a ^13^C signature at 54 ppm which concurs with a –CH_3_ group of a methyl ester (COOCH_3_), and signatures related to the anomeric carbons of the glucopyranose units (C1 at 102 ppm, C4 at 82 ppm), whereas the intense signal at 173 ppm relates to the C6 carbon of the COOH group. The cluster of peaks centered near 70 ppm was assigned to the remaining ^13^C signatures of the glucopyranose units (C2, 3, and 5) of pectin [25,26]. 

In Figure 3c and Figure 4, the ^13^C signal at 173 ppm for the carbonyl group of chitosan (C=O) was shifted to 169 ppm for the PC 51 S composite due to the presence of an acetamide (CONHR; R=acetyl) group. Another intense ^13^C line at 38 ppm was due to the C-H bonds (CH, CH_2_) [22]. These two new signatures in the PC 51 S composite indicate the formation of amide linkage between chitosan and pectin molecules which concurs with a recent report [13]. Besides, a comparison of PC 51 S with unmodified chitosan, C2 peak distorted, and a reduced signal intensity also indicated the formation of a pectin and chitosan composite material.

### 3.4. Particle Size, PDI, and ζ-Potential of Samples

The zeta potential of chitosan is about 17 mV at pH 6.8, as outlined in Table 1, in agreement with another study [24]. At pH values near the pH_pzc_ of chitosan, most amino groups of chitosan are protonated to yield a positive surface charge [27]. The protons on the -COOH groups from the galacturonic acid of the pectin polymers are dissociated at ambient conditions (pH 7), since pH 7 lies above the reported pK_a_ (pectin) ~ 3.5 [28]. The pectin biopolymer shows a net negative surface charge [29], in agreement with the solution conditions (pH > pK_a_, pectin) for the adsorption isotherm. Comparing PC 51 S with PC 51 W composites, the net surface charge of PC 51 S has a more negative ζ-value which may indicate that more -COOH groups are either surface accessible or can be dissociated for this composite (PC 51 S) at pH 6.8, in agreement with the excess pectin content for this material. PC 51 S was prepared under sonication in DMSO that may result in more accessible (less dense) framework for water molecules to access its excess -COOH groups. This feature is in contrast with the more dense PEC framework of PC 51 W, in agreement with the trend in ζ-values in Table 1. More water molecules are available within the PC 51 S framework, as evidenced by the ability to exchange with a cationic dye (methylene blue) [13]. As the pectin content increased from PC 15 S to PC 51 S, the composites show an increasing trend in magnitude of the negative ζ-values, in accordance with the incremental pectin content of such composites. The negative zeta potential of the pectin and chitosan composites indicate that the amount of pectin was sufficient to neutralize all of the amino groups on chitosan at pH 6.8 [27]. As the pectin content in the composites increased, a more negative ζ-value was observed for these amide-based covalent binary composites.

In Table 1, the particle diameter (d) of the composite decreases as the pectin content increases for the composites. This trend concurs with the strong interaction between chitosan and pectin to form smaller particles. A comparison of the composites, PC 51 S with 992 nm diameter and PC 51 W with 1808 nm, where the PEC (PC 51 W) prepared is ca. two times greater in size. The smaller particle size of the PC 51 S composite may be due to the formation of the amide linkage between pectin and chitosan polymers [13]. As the concentration of the chitosan increased, the particle size of composites increased because the swelling behavior of chitosan and water molecules can be absorbed to increase the size of composites [30,31]. The chitosan particle (d = 723 nm) has the smallest diameter when compared to the composites, which concur with estimates from another study [24]. The polydispersity index (PDI) between 0.1 and 0.5 indicates a typical polydispersity of the suspension [32]. Most of the particle size measurements were within this range, which indicate a relatively uniform particle size, in agreement with the range of PDI values.

### 3.5. Sorption Isotherm Results

The following sections describe adsorption isotherm studies of La (III) and Y (III) at ambient conditions (295 K, pH 7).

#### 3.5.1. Uptake of Y (III) by Pectin-Chitosan Binary Composites

In Figure 5, the Langmuir and Freundlich isotherm models were used to study the adsorption isotherm of the Y (III) ions with Pectin-Chitosan composites. In Table 2, the PC 51 S showed the highest value of the K_L_ constant that indicates the strong binding affinity between the PC 51 S and Y (III). From the adj. R-values in Table 2, the Langmuir isotherm model provides a best-fit to the adsorption results over the Freundlich model. The composites can be described as having homogeneous active surface sites for the uptake of Y (III), where the adsorption process conforms to a monolayer adsorption profile. 

The value of Q_m_ is the maximum amount of Y (III) removed by Pectin-Chitosan composites. Both PC 15 S and PC 11 S have similar Y (III) monolayer uptake capacity, but PC 51 S showed slightly greater uptake towards Y (III). The Q_m_ values in Table 2 indicate that the pectin fraction is mainly responsible for adsorption of the Y (III) ions, where such findings are consistent with other results that suggest the -COOH sites of pectin can interact favorably with cation species [3]. The greater overall uptake of Y (III) versus La (III) by the amide-based composites can be related to difference in the charge density of these REEs, in agreement with the Q_m_ values listed in Table 2 and Table 3.

#### 3.5.2. Uptake of La (III) by Pectin-Chitosan Binary Composites 

A similar adsorption trend in uptake was observed for La (III) adsorption by Pectin-Chitosan composites. The Langmuir model provided a better fit to the adsorption results, as compared with the Freundlich model results in Figure 6. The adsorption of La (III) adopts a monolayer adsorption profile due to the homogeneous nature of the high affinity surface sites of the Pectin-Chitosan composites. The adsorption sites are inferred to be the -COOH sites of pectin due to the known high binding affinity of carboxylates with such multivalent cations [33].

The maximum uptake capacity of La (III) in Table 3 for PC 51 S revealed the most favorable adsorption properties among the composites in aqueous solution. Similar to the trends noted for Y (III) described above, the composite with greater pectin content can bind incremental levels of metal ions. A comparison of the Q_m_ values in Table 2 and Table 3 reveal that the composites adsorb more La (III) than Y (III), in parallel agreement with an independent study led by Kusrini [3].

A comparison of the La^3+^ uptake capacity with various types of polyacrylic acid (PAA) grafted resins reveals that such PAA adsorbents display notable uptake that exceed the Q_m_ values listed in Table 2 and Table 3 by up to 10-fold. For example, activated PAN/CNS—70 NFMs used carbon nanospheres with large surface areas to support dense levels of carboxyl groups for high uptake of La^3+^ [34]. Similarly, PAA-grafted adsorbents also showed high La^3+^ uptake capacity [35,36]. A key difference between PAA and pectin-based biopolymers is the much higher density of -COOH groups in the case of PAA (ca. three-fold higher). In turn, we attribute this to differences in the relative binding affinity of PAA over pectin, where PAA is inferred to have enhanced binding affinity due to the greater propensity of “chelate effects” [37]. While pectin-based materials may also display “chelate effects”, the chain branching in such biopolymers is likely to display attenuated effects, in contrast with efficient PAA-based systems (*cf*. Scheme 1 in [37]). To improve the performance of adsorbents from the present study, synthetic modification aimed at increasing the surface area of such biopolymer frameworks via “pillaring effects” [38], along with extensive surface grafting of active functional groups represent potential strategies [39] that will be explored.

## 4. Conclusions

Binary Pectin-Chitosan composites were synthesized and characterized by various complementary methods. The adsorption performance toward Y (III) or La (III) was evaluated with various binary composites, where the adsorption results were well-described by the Langmuir model. The best-fit results reveal that the composite adsorbents had surface sites that are relatively homogeneous in nature. The surface adsorption sites are dominated by the influence of the -COOH groups of pectin that adopt a monolayer adsorption profile for the REEs studied herein. The PC 51 S composite (5:1 pectin–chitosan) displayed the highest La (III) and Y (III) uptake overall, where favorable uptake was attributed to its high pectin content. The FTIR results of the PC 51 S composite indicated that the carboxylate groups of pectin bind effectively with the metal–ion species. This study contributes to the field of hybrid biopolymer assemblies through the design of sustainable composite adsorbents for the controlled uptake of REE cation species from water. We further illustrate that the structure–properties of such composite adsorbents can be tailored according to the relative biopolymer content and the nature of the solvent (DMSO vs. water) media for synthesis to yield covalent frameworks vs. polyelectrolyte complexes [10]. Further work is underway to the explore structure–function relationships in such biopolymer systems through modified materials design strategies to impart enhanced adsorption properties as noted for PAA-based synthetic resins.

## Figures and Tables

**Figure 1 micromachines-12-00478-f001:**
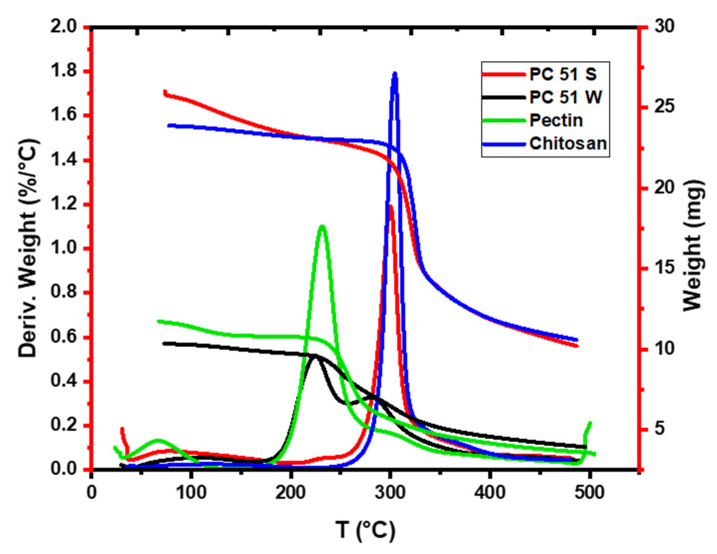
TGA profiles of pectin, chitosan, and the binary composites.

**Figure 2 micromachines-12-00478-f002:**
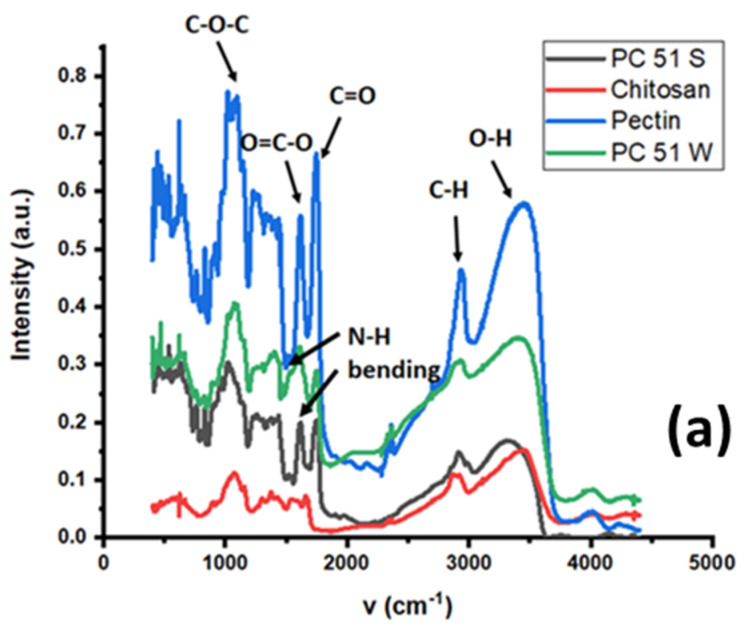
(**a**) FTIR results of pectin, chitosan, and Pectin-Chitosan composites, and (**b**) FTIR results of the PC 51 S composite before and after the La (III) adsorption process.

**Figure 3 micromachines-12-00478-f003:**
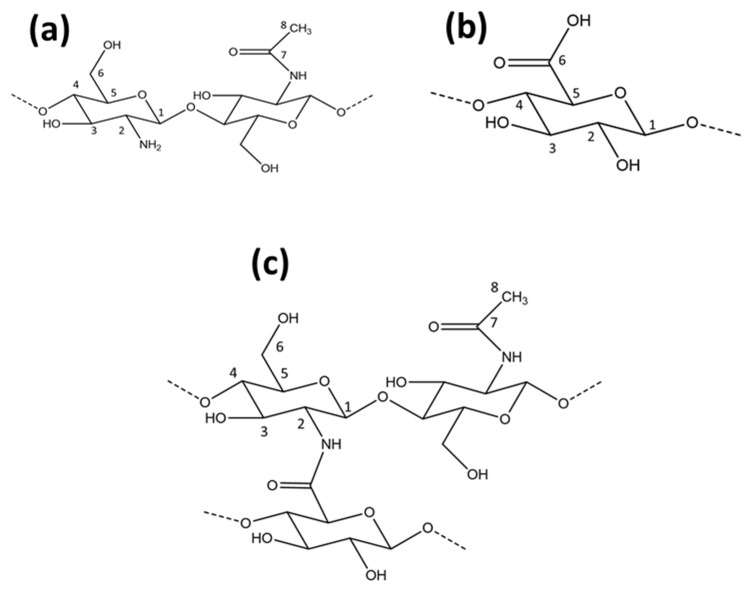
(**a**) The molecular structure of chitosan, (**b**) the molecular structure of pectin, and (**c**) the molecular structure of the amide-based pectin-chitosan covalent framework (PC 51 S).

**Figure 4 micromachines-12-00478-f004:**
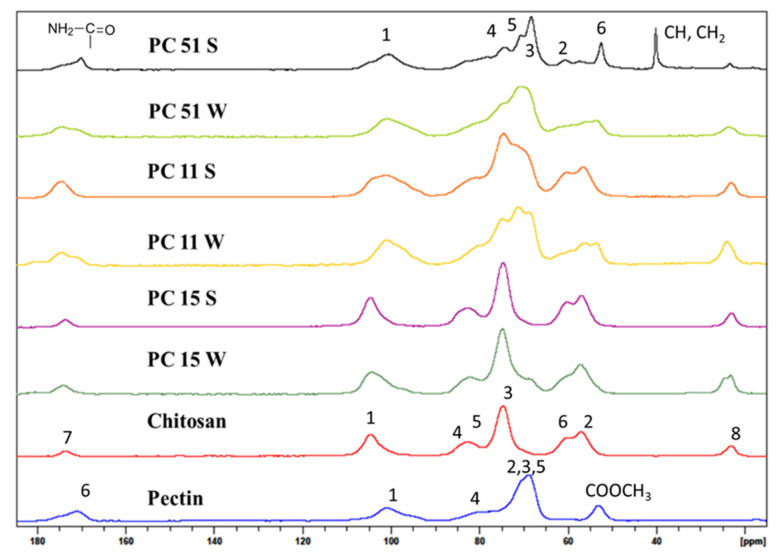
^13^C Solids NMR spectral results for chitosan, pectin, and the binary composites.

**Figure 5 micromachines-12-00478-f005:**
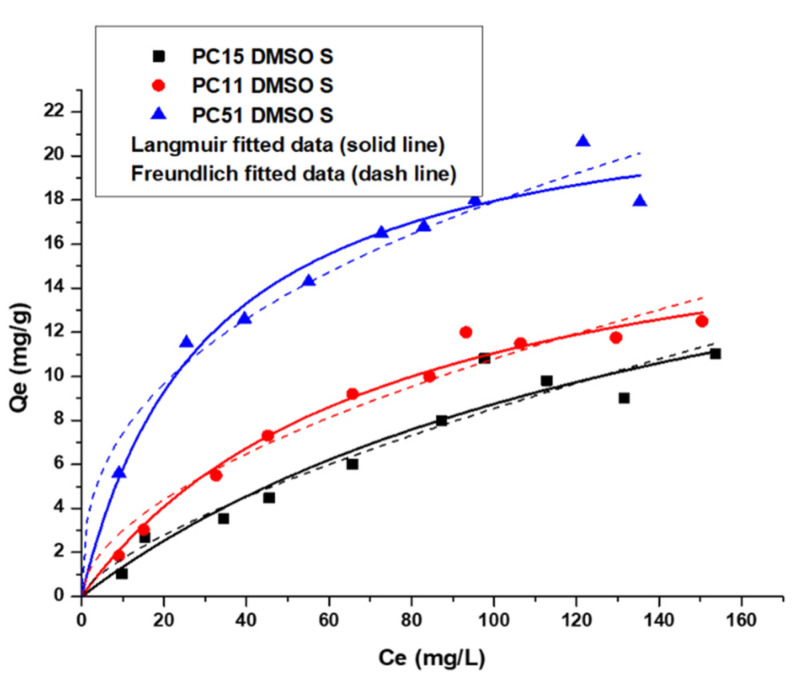
Uptake of Y (III) by binary Pectin-Chitosan binary composites.

**Figure 6 micromachines-12-00478-f006:**
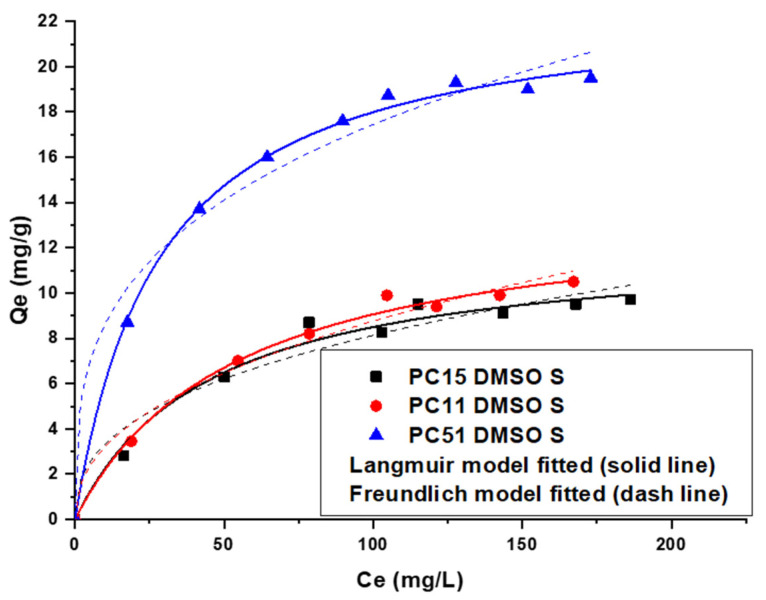
Uptake profile of La (III) by Pectin-Chitosan binary composites.

**Table 1 micromachines-12-00478-t001:** Results of particle size and zeta potential of chitosan and composite samples.

Material	Temp. (°C)	Z-Avg. (d; nm)	PDI	ζ-Value (mV)	Conductivity (mS/cm)
Chitosan	25	723.1	0.636	17.1	0.105
PC 51 S	25	992.0	0.286	−31.4	0.216
PC 51 W	25	1808	0.785	−6.37	0.0581
PC 11 S	25	1300	0.498	−12.2	0.132
PC 15 S	25	1746	0.656	−11.6	0.107

**Table 2 micromachines-12-00478-t002:** Y (III) adsorption parameters for Pectin-Chitosan composites.

*Langmuir model best-fit parameters*			
	K_L_	Q_m_	Adj. R-Square
PC 15 S	0.0063 ± 0.0029	20 ± 6.3	0.94
PC 11 S	0.013 ± 0.0022	19 ± 1.4	0.99
PC 51 S	0.033 ± 0.0060	23 ± 1.3	0.98
*Freundlich model best-fit parameters*			
	K_F_	n	Adj. R-Square
PC 15 S	0.35 ± 0.16	1.4 ± 0.21	0.93
PC 11 S	0.83 ± 0.21	1.8 ± 0.19	0.96
PC 51 S	3.1 ± 0.59	2.6 ± 0.30	0.97

**Table 3 micromachines-12-00478-t003:** La (III) adsorption parameters by Pectin-Chitosan binary composites.

*Langmuir model best-fit parameters*			
	K_L_	Q_m_	Adj. R-Square
PC 15 S	0.022 ± 0.0047	12 ± 0.77	0.97
PC 11 S	0.018 ± 0.0026	14 ± 0.70	0.98
PC 51 S	0.035 ± 0.0024	23 ± 0.40	0.99
*Freundlich model best-fit parameters*			
	K_F_	n	Adj. R-Square
PC 15 S	1.4 ± 0.46	2.6 ± 0.47	0.94
PC 11 S	1.2 ± 0.30	2.3 ± 0.29	0.98
PC 51 S	4.2 ± 0.73	3.3 ± 0.40	0.98

## Data Availability

The data presented in this study are available on request from the corresponding author (L.W.). The data are not publicly available due to the raw/processed data required to reproduce these findings cannot be shared at this time as the data also forms part of an ongoing study.

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
