# Peer review of "Binary Pectin-Chitosan Composites for the Uptake of Lanthanum and Yttrium Species in Aqueous Media"

_micromachines, 2021, doi:10.3390/mi12050478_

Round 1

Reviewer 1 Report

The paper topic is according with the actual trends to replace the synthetic  adsorbents with more sustainable alternative. The natural polymers are one of these alternatives, being biodegradable and non-toxic.

The paper is well structured with adequate abstract, content and conclusions.

Comments:

  1. In this paper is tested the adsorption capacity of pectin-chitosan composites for rare-earth elements such as lanthanum (III) and yttrium(III). These binary composites were obtained in water and in DMSO medium. In paper is not presented if these materials were additionally prepared (i.e. grinding) after filtration and drying.
  2. Besides the adsorption isotherms of Y (III) and La(III) the adsorption efficiency of these two types of composites would have been suggestive.

Author Response

Authors’ Response to Reviewer comments on Manuscript ID: micromachines-1166390

Reviewer #1

The paper topic is according with the actual trends to replace the synthetic  adsorbents with more sustainable alternative. The natural polymers are one of these alternatives, being biodegradable and non-toxic.

The paper is well structured with adequate abstract, content and conclusions.

Comments:

  1. In this paper is tested the adsorption capacity of pectin-chitosan composites for rare-earth elements such as lanthanum (III) and yttrium(III). These binary composites were obtained in water and in DMSO medium. In paper is not presented if these materials were additionally prepared (i.e. grinding) after filtration and drying.

Response: The adsorbents were ground to powder sizes between 45 and 125 µm, and this update is included in the revised manuscript.

  1. Besides the adsorption isotherms of Y (III) and La (III) the adsorption efficiency of these two types of composites would have been suggestive.

Response: The adsorption efficiency of the composites was not compared with the poly acrylic compounds due to the high water solubility of pectin. The water solubility of pectin does not satisfy the phase separation requirements of the sold-liquid adsorption methodology reported herein according to eqn (X). However, for the case of pectin-based biomass such as durian rind, a recent report indicates that La3+ adsorption capacity was 41.2 mg/g pectin (See the following ref; https://doi.org/10.1016/j.jece.2018.10.018). By comparison, a comparable adsorption capacity was observed for binary pectin-activated carbon composites, as follows: La3+ (21.80 mg/g)) and Y3+ (27.78 mg/g); see the following ref; https://doi.org/10.1007/s13369-020-04386-w. The results from these studies are in favourable agreement with the adsorption capacities presented in Tables 2-3 in the manuscript.

In summary, the authors wish to acknowledge Reviewer #1 for the insightful and constructive comments on this manuscript.  We have further edited the work for language, syntax, and clarity throughout to meet the high standards of this journal.

Reviewer 2 Report

This work is that the optimized composition of biopolymer which consists of the pectin and chitosan is synthesized and characterized their structure and thermal stability by various tools to high efficiently capture La and Y ions. However, before acceptance, the following issue must be settled.

1. Can you define whether environmental damages can be caused during fabrication of chitosan composites?

2. In fig1, the yellow line of pectin is not easy to notice. please change another color for readable content.

3. In TGA results, decomposition temperature of 51W is 235 degrees. Please explain that decomposition temperature is too lower than 51W even if it has amide bond same as 51S.

4. In fig4, There are no tests for PC51W and chitosan with La. Please add it to compare with each other.

5. In 3.5.1, Can you describe how much efficient compared to uptake of traditional process with poly acrylic compounds?

Author Response

Authors’ Response to Reviewer comments on Manuscript ID: micromachines-1166390

Reviewer #2:

This work is that the optimized composition of biopolymer which consists of the pectin and chitosan is synthesized and characterized their structure and thermal stability by various tools to high efficiently capture La and Y ions. However, before acceptance, the following issue must be settled.

  1. Can you define whether environmental damages can be caused during fabrication of chitosan composites?

Response: The synthesis of this composite involves DMSO solvent, but this solvent can be recycled safely through solvent recovery methods. The use of sonolysis as part of the synthesis along with renewable feedstocks reduces the energy footprint of the reaction and sustainability of the material inputs, in line with the principles of green chemistry.

  1. In fig1, the yellow line of pectin is not easy to notice. please change another color for readable content.

Response: We agree with the reviewer, the line color was changed to green.

  1. In TGA results, decomposition temperature of 51W is 235 degrees. Please explain that decomposition temperature is too lower than 51W even if it has amide bond same as 51S.

Response: The higher decomposition temperature of 51S is due to the greater covalent character of the composite framework. By comparison, there are fewer amide linkages in 51W due to the role of water in reducing the yield of amide cross-linking. The latter is supported by the relative overlap of the decomposition at 235 deg. C that coincides with that of pristine pectin (see the green line in Fig. 1).

  1. In fig4, There are no tests for PC51W and chitosan with La. Please add it to compare with each other.

Response: In a previous publication (J. Compos. Sci. 2020, 4, 95; doi:10.3390/jcs4030095), PC 51 W was shown to have a lower uptake of a cationic dye (methylene blue) than PC51 S (DMSO). We have selected the composites prepared in DMSO as the adsorbent of choice due to their greater structural stability in water. The greater stability of PC51 S is in agreement with the amide linkages that stabilize the composite framework. By comparison, pristine chitosan has limited uptake capacity with cation species, in comparison with composites such as PC51 S.  This is related to the greater porosity and accessibility of adsorption sites of cross-linked composites. This is in agreement with the lower surface area of chitosan due to its semi-crystalline morphology which has been well documented in literature (J. Colloid Interface Sci. 2012, 388, 225–234).

  1. In 3.5.1, Can you describe how much efficient compared to uptake of traditional process with poly acrylic compounds?

Response: The adsorption efficiency was compared with poly acrylic compounds from the literature in the final paragraph preceding Section 4. In general, the PAA-based composites show markedly higher adsorption capacity relative to pectin-based composites.  We attribute this difference due to the higher density of carboxylate groups in the case of PAA and the potential role of chelation. The role of chelation is inferred to be less pronounced in pectin materials, in agreement with the adsorption capacity results reported herein. Notwithstanding the above comments, pectin-based materials are anticipated to contribute to the field of sustainable adsorbents according to the tunability of their structure and physicochemical properties, as reported herein.

In summary, the authors wish to acknowledge Reviewer #2 for the insightful and constructive comments on this manuscript.  We have further edited the work for language, syntax, and clarity throughout to meet the high standards of this journal.

Round 2

Reviewer 2 Report

The authors addressed the comments given in the previous review, and I would recommend publishing them.